# The Effect of *Ferula communis* Extract in *Escherichia coli* Lipopolysaccharide-Induced Neuroinflammation in Cultured Neurons and Oligodendrocytes

**DOI:** 10.3390/ijms22157910

**Published:** 2021-07-24

**Authors:** Jessica Maiuolo, Irene Bava, Cristina Carresi, Micaela Gliozzi, Vincenzo Musolino, Miriam Scicchitano, Roberta Macri, Francesca Oppedisano, Federica Scarano, Maria Caterina Zito, Francesca Bosco, Stefano Ruga, Saverio Nucera, Sara Ilari, Ernesto Palma, Carolina Muscoli, Vincenzo Mollace

**Affiliations:** IRC-FSH Department of Health Sciences, University “Magna Græcia” of Catanzaro, 88100 Catanzaro, Italy; irenebava@libero.it (I.B.); carresi@unicz.it (C.C.); micaela.gliozzi@gmail.com (M.G.); xabaras3@hotmail.com (V.M.); miriam.scicchitano@hotmail.it (M.S.); robertamacri85@gmail.com (R.M.); oppedisanof@libero.it (F.O.); federicascar87@gmail.com (F.S.); mariacaterina.zito@libero.it (M.C.Z.); boscofrancesca.bf@libero.it (F.B.); rugast1@gmail.com (S.R.); saverio.nucera@hotmail.it (S.N.); sara.ilari@hotmail.it (S.I.); palma@unicz.it (E.P.); muscoli@unicz.it (C.M.); mollace@libero.it (V.M.)

**Keywords:** ferutinin, oligodendrocytes, neurons, inflammation, demyelinating pathologies, oxidative stress

## Abstract

In recent decades, interest in natural compounds has increased exponentially due to their numerous beneficial properties in the treatment of various acute and chronic diseases. A group of plant derivatives with great scientific interest is terpenic compounds. Among the plants richest in terpenes, the genus *Ferula* L. is one of the most representative, and ferutinin, the most common sesquiterpene, is extracted from the leaves, rhizome, and roots of this plant. As reported in the scientific literature, ferutinin possesses antioxidant and anti-inflammatory properties, as well as valuable estrogenic properties. Neurodegenerative and demyelinating diseases are devastating conditions for which a definite cure has not yet been established. The mechanisms involved in these diseases are still poorly understood, and oxidative stress is considered to be both a key modulator and a common denominator. In the proposed experimental system, co-cultured human neurons (SH-SY5Y) and human oligodendrocytes (MO3.13) were treated with the pro-inflammatory agent lipopolysaccharide at a concentration of 1 μg/mL for 24 h or pretreated with ferutinin (33 nM) for 24 h and subsequently exposed to lipopolysaccharide 1 μg/mL for 24 h. Further studies would, however, be needed to establish whether this natural compound can be used as a support strategy in pathologies characterized by progressive inflammation and oxidative stress phenomena.

## 1. Introduction

In recent decades, scientific interest in plant derivatives has increased exponentially due to their innumerable beneficial activities in different areas of the body and against various acute and chronic diseases. Indeed, most natural compounds have anti-inflammatory, antioxidant, and anti-cancer properties, among others [1,2]. Terpenic compounds are a group of plant derivatives of great scientific interest. These compounds can be extracted from a wide range of plants, and their basic chemical structure consists of a number of repeated isoprene (C_5_H_8_) units [3,4]. The main terpenic compounds are monoterpenes (C10), sesquiterpenes (C15), diterpenes (C20), triterpenes (C30), and tetraterpenes (C40) [5]. The most beneficial effects of terpenic compounds are their antioxidant properties [6], although their use as penetration enhancers for transdermal drug delivery has recently become widespread [7]. In particular, sesquiterpenes [8,9] have shown interesting properties in the protection of health, such as anti-inflammatory [10], anticancer [11,12], and bactericidal activities [13]. However, sesquiterpenes notably demonstrate a dose-dependent effect, which means that low concentrations tend to be protective while higher doses are toxic [14]. This double effect is caused by the permeability of the biological membranes to cationic species, which, in a dose-dependent way, undergo modifications on a part of the sesquiterpenes [15,16]. Among the plants richest in sesquiterpenes, the genus *Ferula* L. is one of the best known [17,18,19]. The genus *Ferula* L. consists of about 170 species that mainly grow in Northern Africa, Mediterranean areas, and Central Asia [20]. In Italy, three species have been described: *Ferula communis* L., *Ferula arrigonii* Bocchieri, and *Ferula glauca* L. Notably, some plants of the genus *Ferula* have been used as pharmaceutical plants for many decades. *Ferula* is a rich source of biologically active compounds, such as sesquiterpenes, coumarin derivatives, disulfide compounds, and aromatic lactones. Extracts and metabolites from *Ferula* possess important biological properties, including anti-inflammatory [21], antiviral [22], anticancerous [23], antidiabetic [24], and anti-bacterial [25] properties. The most common sesquiterpene in non-toxic chemotypes of *Ferula* L. is ferutinin (FER), which is commonly extracted from the roots, leaves, and rhizomes of plants [26]. Both in vitro and in vivo, FER has been shown to possess valuable estrogenic properties in the scientific literature [27,28]. Some structural studies confirmed the estrogenic properties of this natural compound, justified by the correct distance between the oxygen of the alcohol hydroxyl group and the methoxy group in the benzene ring [29]. In this study, we evaluated the effects of FER extracted from *Ferula communis* L. in an in vitro model of co-cultures of human neurons and oligodendrocytes exposed to lipopolysaccharide (LPS), a heat-stable cell-wall component of Gram-negative bacteria. LPS determines the induction of a systemic inflammatory response that can lead to cell damage, multiple organ failure, and shock [30]. LPS stimulates the overproduction of pro-inflammatory mediators and cytokines, including Interleukin-1β, Interleukin 6, Tumor Necrosis Factor, prostaglandins, Interferon-γ, and Reactive C protein, resulting in an acute inflammatory response [31]. In addition, LPS determines mitochondrial dysfunction by facilitating the production of Reactive Oxygen Species (ROS), which are particularly toxic to the cell. Moreover, it is well known that ROS govern the LPS-induced pro-inflammatory response in neuroglia cells via several pathways [32] and significantly contribute to the deterioration of neuronal cells. They do this by modulating the functionality of different biomolecules (lipids, proteins, DNA, and RNA) [33]. Since the brain consumes a large amount of oxygen to ensure its proper functioning and produces a large amount of ROS, this organ can be considered the strategic point of neurodegeneration [34]. Neurodegenerative diseases are devastating conditions for which a cure has not yet been defined. The mechanisms involved in such diseases are still poorly understood, and oxidative stress is considered to be both a key modulator and a common denominator. For this reason, we will investigate the effects of FER on a known model of oxidative stress to evaluate some aspects that translate into diseases that afflict the nervous system.

## 2. Results

### 2.1. Both MO3.13 and SH-SY5Y Express Estrogen Receptor-β (ER-β)

The SH-SY5Y and MO3.13 cell lines express ER-β, as already evidenced in the scientific literature [35,36,37,38]. As shown in Figure 1, this fundamental information was confirmed in our experimental model of co-cultures. The presence of the receptor was determined via RT-PCR (Figure 1, panel a) by means of western blotting (Figure 1, panel b) and cytometry (Figure 1, panel c). Notably, there was no difference between the two cell lines, and the receptor was equally expressed.

### 2.2. 17-Beta Estradiol (17-β-E2) Protects Oligodendrocytes and Neurons from LPS-Induced Damage

First, preliminary experiments were performed to evaluate the percentage of cell death induced by exposing the co-cultured SH-SY5Y and MO3.13 cell lines to LPS. A concentration of 1 μg/mL for 24 h, in accordance with the scientific literature, was chosen, and all experiments were performed using this dosage. Treatment with LPS caused notable cell damage accompanied by reduced viability (*p* < 0,01 and *p* < 0.001 in neurons and oligodendrocytes, respectively), which can be seen in Figure 2, which shows this effect on both cell lines—slightly more so in the oligodendrocytes than in the neurons. In experiments involving 17-β-E2, the compound was applied to SH-SY5Y and MO3.13 cells 1 h before the addition of LPS and was present during the entire exposure period of 24 h. The chosen concentration of 17-β-E2 was 1 μM according to the published literature [37]. No changes in the viability, neurons, or oligodendrocytes were noted when 17-β-E2 alone was used. Conversely, co-treatment with LPS showed that 17-β-E2 reduced LPS-induced mortality in both cell lines, with statistically significant protection.

### 2.3. Effects Induced by Treatment with FER

Since it is known that FER exerts estrogenic actions [27,28], we sought to assess whether FER would present the same behavior as 17-β-E2 in our experimental system. First, we constructed a dose–response curve in which the neurons and oligodendrocytes were exposed to increasing concentrations of FER for 24 h. The results showed a dose-dependent effect: Lower concentrations of FER (1–33 nM) did not lead to cell suffering or death, excluding toxic effects. Conversely, higher concentrations (66–500 nM) showed a gradual and increasing toxicity that resulted in a reduction of cell viability. This result was similar for both SH-SY5Y and MO3.13 cells. To select a concentration of FER for further experiments, we chose a higher and non-toxic dose of 33 nM (the corresponding data are shown in Figure 3, panel a). Subsequently, we sought to assess whether the damage induced by LPS could be reversed by FER, as is the case for 17-β-E2. The results for cell viability indicated that FER was able to restore LPS-induced damage significantly in both cell lines (as shown in Figure 3, panel b). LPS presumably caused apoptotic damage, as shown by the modulation of the expression of the cleaved caspase-3, confirming the mechanism of action exerted by LPS. Once again, pretreatment with FER protected both cell lines from the activation of caspase-3 (*p*-value < 0.05 vs. LPS in neurons; *p*-value < 0.01 vs. LPS in oligodendrocytes; see Figure 3, panels c,d).

### 2.4. Evaluation of Cell Death through an Annexin V-PI Assay

To determine whether the LPS-induced cell death was apoptotic or necrotic, the cells were stained with Annexin V-FITC and PI, and the study was carried out using flow cytometry. As shown in Figure 4, the FER did not indicate any cell suffering, and the cells were Annexin V-negative/PI-negative. The cells treated with LPS were Annexin V-positive/PI-negative, indicating apoptotic death; treatment with LPS also showed a minimal amount of V-positive/PI-positive cells, suggesting that post-apoptotic secondary necrosis was rarely induced. The cells co-treated with LPS + FER showed greater viability than those treated with LPS, indicating that the natural compound limited induction of the apoptotic process. However, this effect was more evident in the neurons than in the oligodendrocytes. Relative quantification is shown in Figure 4, panel b.

### 2.5. Oxidative Profile in SH-SY5Y and MO3.13 Cells

To investigate the potential protective effects of FER in our experimental system, we first measured the intracellular ROS and MDA content after treatment with LPS. Figure 5 shows the results obtained from the ROS determination in both the SH-SY5Y and MO3.13 cells using a flow cytometric assay. It is known that LPS produces ROS [32]. The results of the current study confirm these reports, as the intracellular ROS levels significantly increased when both cell lines were treated with LPS. A significantly higher level of ROS (*p*-value < 0.01; *p*-value < 0.001 in SH-SY5Y and MO3.13 respectively) was detected in the cells treated with LPS compared to the control. In addition, cell pretreatment with FER followed by LPS significantly reduced ROS accumulation. In both cell lines, H_2_O_2_ was used as the positive control. Subsequently, we measured the levels of the biomarker MDA to assess whether treatment with LPS could lead to lipid peroxidation reactions. As demonstrated in the scientific literature [39], LPS generated an increase in MDA content in both cell lines. However, among the oligodendrocytes, the damage was greater than that among the neurons. Moreover, in this case, pretreatment with FER led to a reduced formation of MDA, evidencing the ability of this natural compound to safeguard the lipidic structure of the membranes, in addition to the structures of the other biological molecules (see Figure 6, panels a,b).

### 2.6. Determination of Antioxidative Status in SH-SY5Y and MO3.13 Cells

Catalase (CAT), superoxide dismutase (SOD), and glutathione peroxidase (GSH-Px) are three of the primary antioxidant enzymes contained in mammalian cells and are of fundamental importance to the survival of organisms undergoing oxidative stress [40]. The most important parameter determining the biological impact of antioxidant enzymes is activity. In our experimental model, we measured the activity of CAT, SOD, and GSH-Px following several rounds of treatment with LPS (1, 6, 9, 18, and 24 h), enabling us to construct a curve of time-dependent enzymatic activity. As shown in Figure 6, panels c,d, the activity of CAT was reduced after 9 h of treatment with LPS and appeared statistically significant starting at 9 h of treatment. Moreover, the oligodendrocytes were more involved than the neurons. A reduction in SOD activity occurred after 6 and 9 h of LPS treatment in the oligodendrocytes and neurons, respectively, and, similar to the activity of CAT, the MO3.13 cells were more vulnerable than the SH-SY5Y cells (see Figure 7, panels a,b). Finally, as shown in Figure 7, panels c,d), the activity of the GSH-Px enzyme was the same as that of CAT and SOD, with an observable reduction after 6–9 h treatment with LPS, at which point the oligodendrocytes showed a greater decrease than the neurons. Notably, for all three enzymes, pretreatment with FER was able to improve the enzymatic activity that was reduced by LPS.

## 3. Discussion and Future Perspectives

Inflammation is the common denominator of multiple neurodegenerative diseases, including Alzheimer’s disease, Parkinson’s disease, Multiple sclerosis, Amyotrophic lateral sclerosis, Huntington’s disease, and frontotemporal dementia [41,42]. Today, it is known that the brains of patients with neurodegenerative diseases are characterized by activation of the microglia, elevated levels of pro-inflammatory cytokines, and marked astrocytosis [43]. Based on this knowledge, it is essential to reduce or eliminate the inflammatory processes in these diseases. For this reason, drugs and natural products with anti-inflammatory properties have been evaluated in animal models of neurodegeneration and neuroinflammation [44,45]. The involvement of inflammation also occurs in demyelinating diseases in which the oligodendrocytes are more vulnerable [46,47]. LPS has been used worldwide in experimental in vitro and in vivo models of neuroinflammation [48,49], including multiple sclerosis [50]. LPS is a potent endotoxic component of the outer membranes of Gram-negative bacteria. Due to its high resistance to degradation, LPS remains in the organism for a long period of time, providing a persistent inflammatory stimulus and producing proinflammatory cytokines that activate the neuroimmune system. The induced excessive inflammatory response also provokes mitochondrial dysfunction, a reduction in ATP synthesis, the accumulation of ROS, and damage to the systemic vascular endothelium [51,52]. For these reasons, in this experimental work, we used LPS as an inflammatory stimulus in an in vitro model of co-cultured human neurons and oligodendrocytes. Our first goal was to induce an inflammatory condition that reproduced that of the main neurodegenerative diseases involving physiological cross-talk between neurons and oligodendrocytes grown in direct contact with each other. The results showed that LPS-induced damage involved more oligodendrocytes than neurons, demonstrating the greater fragility of this cell line. This deduction was confirmed by the results concerning the reduction of cell viability (Figure 2, panel a), the accumulation of ROS (Figure 5, panels a–c) and the biomarker MDA (Figure 6, panels a,b), and the evaluation of the antioxidative status performed by measuring the enzymatic activity of CAT, SOD, and GSH-PX (Figure 6, panel c; Figure 7, panels a–d). It is known that oligodendrocytes provide metabolic support to neurons. Axons, for example, are often very long and require significant amounts of energy to carry out their functions properly. Since neurons have no significant energy reserves, oligodendrocytes have evolved to meet the energy demands of neurons [53]. It is also well-known that oligodendrocytes are a primary target in certain neurodegenerative diseases, such as multiple sclerosis, a chronic neuroinflammatory disease characterized by permanent inflammation that generates oligodendrocyte damage and the demyelination of axons. However, since the oligodendrocytes are compromised in MS, and the axons lose their functional myelin sheathes, it is assumed that the metabolic support of the oligodendrocytes would fail under inflammatory conditions and that the oligodendrocytes would become more vulnerable than the neurons. Indeed, in a mouse model of Experimental Autoimmune Encephalomyelitis (EAE), axonal damage occurred before axonal demyelination, suggesting that the loss of the metabolic support of the oligodendrocytes may be a very early event in the disease, which would confirm the specific vulnerability of the oligodendrocytes [54]. Nevertheless, further studies are needed to confirm this hypothesis. The regulation of neuroinflammation is, therefore, essential and could represent a potential strategy to alleviate the symptoms in neurodegenerative diseases.

Estrogens play a known neuroprotective role, as confirmed in many models of Alzheimer’s disease, Parkinson’s disease, and multiple sclerosis. In a previous study, ovariectomies in rodents were clearly associated with the up-regulation of a large number of inflammatory markers [55]. The augmented expression of inflammatory markers was also observed in postmenopausal women [56]. Estrogens have shown anti-inflammatory, antioxidant, and anti-apoptotic properties that improve cognitive performance [57]. Since it was previously demonstrated that estrogen and estrogen receptor agonists inhibit dysfunction and cell death in the nervous system [58], the second objective of this scientific work was to evaluate the effects of FER on a model of inflammation induced by LPS in co-cultured oligodendrocytes and neurons. In the literature, sesquiterpene FER extracted from *Ferula communis* L. was shown, both in vitro and in vivo, to possess valuable estrogenic properties [37,38,59,60]. In the present study, we first evaluated whether the SH-SY5Y and MO3.13 cell lines expressed estrogenic receptors, as reported in the literature [61,62]. As illustrated in Figure 1, both the neurons and oligodendrocytes possess genetic information and can express ER-β. In this study, attention was focused on ER-β, while no experiments were conducted on ER-α. Figure 2, panel b confirms what was already published regarding the protection induced by estrogens and shows how co-treatment with FER-LPS can significantly protect against the damage induced by LPS. After confirming this basic information, we constructed a dose–response curve of FER for both cell lines to select a concentration of FER that could be used in subsequent experiments (33 nM). Treatment with FER did not cause any toxic effect. However, when we co-treated both cell lines with FER and LPS, we always evaluated the reversal or reduction of any inflammatory damage induced by LPS. The protective effects of FER against LPS were demonstrated in terms of cell viability (Figure 3, panel b) through the activation of cleaved caspasi-3 (Figure 3, panels c,d), in the description of apoptotic death (Figure 4, panels a,b), and in the evaluation of the oxidative profile and antioxidative status. It is known that ROS significantly contribute to the death of neuronal cells and modulate the functions of important biomolecules, such as lipids, proteins, DNA, and RNA [33,63]. Similarly, oligodendrocytes appear to be particularly vulnerable to ROS and not only induce cell death but also inhibit differentiation from progenitor oligodendrocytes to mature oligodendrocytes [64]. Evaluating the oxidative profiles and antioxidative status constituted the third fundamental objective of this study. These evaluations were performed to (a) determine whether FER protects neurons and oligodendrocytes from oxidative damage induced by LPS and (b) assess whether oligodendrocytes can support neurons by activating the enzyme activity of the three main antioxidant enzymes, CAT, SOD, and GSH-Px, under our experimental conditions. The results were as follows:(a)In this experimental system, neurons and oligodendrocytes were subjected to the accumulation of LPS-induced ROS and MDA, and FER was able to protect these cell lines from both;(b)The enzymatic activity of CAT, SOD, and GSH-Px was reduced after only 6 h of treatment with LPS, and the antioxidative status of the oligodendrocytes appeared to be particularly damaged. Presumably, therefore, neurons are not supported by oligodendrocytes in the management of antioxidative status. In any case, FER-LPS co-treatment was shown to support neurons and oligodendrocytes by protecting them from oxidative insult.

In light of these results, the present study provided three main conclusions:Co-cultured Oligodendrocytes and neurons were a good experimental model to investigate because of the cross talk that physiologically exists between these cell lines. In particular, oligodendrocytes were found to be more vulnerable than neurons; oligodendrocytes were able to support neurons metabolically but were fragile from the perspective of antioxidative status;Pretreatment with FER was highly protective against the damage caused by LPS in both cell lines. These results highlight the protective action of estrogen, since FER exerts estrogenic actions, and indicate the use of FER as a potential therapeutic strategy to reduce the damage caused to neurons and oligodendrocytes in inflammatory neurodegenerative diseases. Thus, for the first time, we can include the use of sesquiterpene FER in the scenario of neurodegeneration. Finally, in this direction, FER could be an excellent candidate to replace the neuro-protective action of estrogen in menopause;Under inflammatory conditions, the antioxidative profiles of oligodendrocytes are greatly compromised. This information could, therefore, be of prime importance in the treatment of demyelinating diseases. Indeed, the protective role of natural compounds in neuroinflammatory diseases is becoming an important topic in scientific research [65,66,67].

The cellular mechanism by which FER was able to exert a protective effect in this experimental model was presumably the high amount of polyphenols in the extract considered. For this reason, it would be interesting to identify the most widely represented components to test them individually. Continuation of this work could involve two important phases: firstly, chemical analyses carried out with HPLC could provide the detailed composition of the FER extract; secondly, further experiments could be carried out (in vitro and in vivo) using the most widely represented components found. Through these studies, we could determine if a single component or more components are responsible for the effects generated, thereby offering important information for preclinical and clinical medicine.

## 4. Materials and Methods

### 4.1. LPS and Ferutinin

LPS was purchased from Sigma Aldrich, 20151 Milan, Italy.

FER was kindly provided by Herbal and Antioxidant Derivatives S.r.l. (Polistena, RC, Italy).

### 4.2. Cell Cultures

The undifferentiated human oligodendrocyte cell line (MO3.13) and human neuronal line (SH-SY5Y) were purchased from the American Type Culture Collection (20099 Sesto San Giovanni, Milan, Italy). Before carrying out the treatments, the neurons and oligodendrocytes were suitably differentiated. SH-SY5Y differentiation was induced using 10 μM of all-trans retinoid acid (Sigma Aldrich, 20151 Milan, Italy) for 5 days. Since the MO3.13 oligodendrocytes are a hybrid resulting from a combination of human rhabdomyosarcoma cells and adult human oligodendrocytes, the cells were differentiated in mature oligodendrocytes via treatment with Phorbol 12-myristate 13-acetate (Sigma Aldrich, 20151 Milan, Italy) (100 nM) for 5 days [29]. All the substance concentrations were carefully evaluated and based on the published literature. Following differentiation, the experiments were carried out using cell lines grown in a co-culture in 12-well Transwell insert plates. These specific plates were selected based on the presence of a polyester membrane with 1 µm pores to prevent cell migration. To this end, the differentiated oligodendrocytes and neurons were placed, respectively, on the outer and the inner portions of the Transwell inserts. Under these experimental conditions, only the growth medium came into contact with both the cell lines. To better understand the experimental model description, please see [29]. Both cell lines were cultured in Dulbecco’s modified Eagle’s medium (DMEM) reinforced with 10% fetal bovine serum (FBS), 100 U/mL penicillin, and 100 µg/mL streptomycin in a humidified 5% CO_2_ atmosphere at 37 °C. The medium was changed every 2–3 days, and when the cell lines reached 50% confluence, they were treated with FER 33 nM for 24 h. At the end of this treatment, the cells were exposed to 1 μg/mL LPS for 24 h, and then the appropriate tests were carried out.

### 4.3. Cell Viability

The MTT test is based on the observation that live cells with active mitochondria are able to reduce 3-(4,5-dimethylthiazol-2-yl)-2,5-diphenyltetrazolium bromide (MTT) to a dark blue visible reaction product. This reaction is used to evaluate cell viability. The SH-SY5Y and MO3.13 cells were placed in 96-well microplates at a density of 6 × 10^3^ and, the next day, were treated with FER as indicated. At the end of the treatment, the cells were exposed to LPS for 24 h. Subsequently, the medium was replaced with a phenol red-free medium containing MTT solution (0.5 mg/mL) and, after 4 h incubation, 100 μL of 10% SDS was added to each well to solubilize the formazan crystals. The microplates were then gently agitated, and the optical density was measured at wavelengths of 540 and 690 nm using a spectrophotometer (X MARK Spectrophotometer Microplate Bio-Rad). The results were expressed as the percentages of untreated cells and used to calculate the relative cell viability.

### 4.4. Cell Lysis and Immunoblot Analysis

Cell monolayers in 100 mm plates were washed with ice-cold PBS and lysed with a pre-heated (80 °C) lysis buffer containing 50 mM Tris-HCl (pH = 6.8), 2% SDS, and a protease inhibitor mixture and immediately boiled for 2 min. The protein concentration in the cell lysates was determined using a DCA protein assay. After the addition of 0.05% bromophenol blue, 10% glycerol, and 2% β mercaptoethanol, the samples were boiled again and loaded into SDS-polyacrylamide gels (12%). Following electrophoresis, the polypeptides were transferred to nitrocellulose filters, blocked with TTBS/milk (TBS 1%, Tween 20, and non-fat dry milk 5%), and then the antibodies were used to reveal the respective antigens. Primary antibodies were incubated overnight at 4 °C followed by a horseradish peroxidase-conjugated secondary antibody for 1 h at room temperature. The blots were developed using the chemiluminescence procedure. The following primary antibodies were used: a mouse monoclonal antibody for Estrogen Receptor beta (Abcam, ERb455-ab187291) at 1:1000 dilution, a mouse monoclonal antibody for Cleaved Caspase-3 (Abcam, E83-77-ab32042) at 1:1000 dilution, and a mouse monoclonal anti-actin antibody (Sigma Aldrich) at 1:5000 dilution. Horseradish peroxidase-conjugated goat anti-mouse antibody was used as the secondary antibody at 1:10.000 dilution. For the flow cytometry protocol, the cells were placed in ice cold PBS, 10% FCS, and 1% sodium azide. The cells were then added to a suspension of 1–5 × 10^6^ cells/mL, and 2 ug/mL of the primary antibody, and the dilution was performed in 3% BSA/PBS for 30 min. The cells were then incubated with a secondary antibody at room temperature in the dark and analyzed immediately using a flow cytometer (Becton Dickinson, Milan, Italy).

### 4.5. Intracellular ROS Detection

ROS identification was based on oxidation of the permeable non-fluorescent probe H_2_DCF-DA. This is because H_2_DCF-DA readily diffuses into the cells where intracellular esterases cleave the acetate group of H_2_DCF-DA from the molecule to yield H_2_DCF, which is entrapped within the cells. Intracellular ROS oxidizes H_2_DCF to form the highly fluorescent compound DCF. The cell lines were plated in 96-well microplates at a density of 6 × 10^4^ and were treated the following day as described. At the end of the treatment, the growth medium was replaced with a fresh medium containing H_2_DCF-DA (25 μM). After 30 min at 37 °C, the cells were washed twice to remove the extracellular H_2_DCF-DA and centrifuged and resuspended in PBS. Then, in the presence of H_2_O_2_ (or not) (100 μM, 30 min of incubation), the fluorescence was evaluated using flow cytometry analysis. A total event of 10,000 cells per sample was acquired using a FACS Accuri laser flow cytometer (Becton Dickinson, Milan, Italy).

### 4.6. Malondialdehyde Assay

Lipid peroxidation was determined by measuring the results of the reaction of malondialdehyde (MDA) with thiobarbituric acid to form a colorimetric product, proportional to the MDA present. The MDA levels were, therefore, considered to be an index of lipid peroxidation. The cell cultures were plated and treated the following day, as indicated. At the end of the treatment, the cells were scraped. The cell suspension underwent a freeze/thaw cycle, and a mixture consisting of 36 mM TBA solubilized in glacial acetic acid was added and heated for 60 min at 100 °C. The reaction was halted by placing the vials in an ice bath for 10 min, and the absorbance was measured by means of a spectrometer (Thermo Fisher Scientific Inc., Milan, Italy) at 532 nm.

### 4.7. Annexin V Staining

The cells were treated as indicated above. Then, the cells were detached using trypsin, washed twice with cold PBS, and suspended in 1× binding buffer (Metabolic Activity/AnnexinV/Dead Cell Apoptosis Kit) at a concentration of 1 × 10^6^ cells/mL. One hundred microliters of the suspension were transferred to a 5 mL culture tube, and 5 μL of FITC Annexin V (BD Biosciences, San Jose, CA, USA) was added. The samples were gently vortexed and incubated for 15 min at 25 °C in the darkness. Finally, 400 μL of 1× Binding Buffer and 5 μL propidium iodide (PI) were added to each tube, and the samples were analyzed by flow cytometry over 1 h (emission filters of 515–545 nm for FITC and 600 nm for PI). A total event of 20,000 cells per sample was acquired, and the fluorescence was evaluated using FACS Accuri laser flow cytometry (Becton Dickinson, Milan, Italy).

### 4.8. Total RNA Extraction and RT-PCR

Total RNA was extracted using an RNeasy Mini Kit along with QIAshredder (Qiagen, Milan, Italy) according to the manufacturer’s instructions. Any contaminating DNA present in the sample was degraded via in-column incubation with DNaseI (Qiagen, Milan, Italy) for 15 min. The amount of eluted total RNA was determined spectrophotometrically at 260 nm, and its purity was evaluated using a 260:280 ratio. One microgram of each sample was reverse-transcribed using the SuperScript III FirstStrand Synthesis System for RT-PCR (Invitrogen, Milan, Italy). Primers for detecting fragments of the Estrogenic Receptor- β (ER-β) gene were designed from a published human sequence [35]. Specifically, the primers were 59-GGCCGACAAGGAGTTGGTA-39 (nucleotides 762–780) and 59-AAACCTTGAAGTAGTTGCCAGGAGC-39 (nucleotides 995–1020), yielding an amplified product of 257 bp. The PCR reaction contained 2 units of DNA polymerase BIOTAQ; 10X PCR buffer (containing 1.5 mM MgCl_2_; both from Bioline, London, United Kingdom); 0.5 mg of each oligonucleotide primer; 200 μM each of dATP, dCTP, dGTP, and dTTP; 2 μL of nascent cDNA; and sterile distilled water to bring the volume to 50 μL. The PCR products (20 μL) were resolved in 1.2% agarose gel in a Tris-borate-EDTA buffer and visualized via ethidium bromide staining under UV illumination [36].

### 4.9. Superoxide Dismutase (SOD), Glutathione Peroxidase (GSH-Px), and Catalase (CAT) Activities

Following the aforementioned treatments, both cell lines were collected to observe the activities of SOD, GSH-Px, and CAT using the relative kits (Abcam, Cambridge, United Kingdom), according to the manufacturer’s instructions. In particular, three SOD enzymes were highly compartmentalized, and the SODs converted the superoxide radicals into hydrogen peroxide and molecular oxygen (O_2_). Copper- and zinc-containing superoxide dismutase (CuZnSOD) is located in the cytoplasm and represents approximately 90% of the total SOD activity in a eukaryotic cell. To determine SOD activity using a test, superoxide anions react with a specific probe to produce a water-soluble formazan dye that can be detected by an increase in absorbance to 450 nm. The higher the SOD activity in the sample, the lower the formazan dye produced. GSH-Px converts hydrogen peroxide into water and, in a test to discourage GSH-Px activity, Gpx oxidizes glutathione to produce glutathione disulfide (GSSG) as part of the reaction in which Gpx reduces mercury hydroperoxide. Glutathione reductase (GR) reduces GSSG to produce GSH and, in the same reaction, consumes NADPH. The decrease in NADPH was measured at 340 nm and was proportional to Gpx activity. CAT then converts hydrogen peroxide into oxygen and water. In the assay to determine CAT activity, the unconverted H O reacts with the Oxired probe to produce a product measurable at 570 nm whose activity is inversely proportional to the signal. GR reduces the GSSG to produce GSH and, in the same reaction, consumes NADPH. The decrease in NADPH was measured at 340 nm and was proportional to Gpx activity.

### 4.10. Statistical Analysis

Data were expressed as the mean ± standard deviation (SD) and statistically evaluated for differences using a one-way analysis of variance (ANOVA) followed by a Tukey–Kramer multiple comparison test (GraphPad software for science).

## Figures and Tables

**Figure 1 ijms-22-07910-f001:**
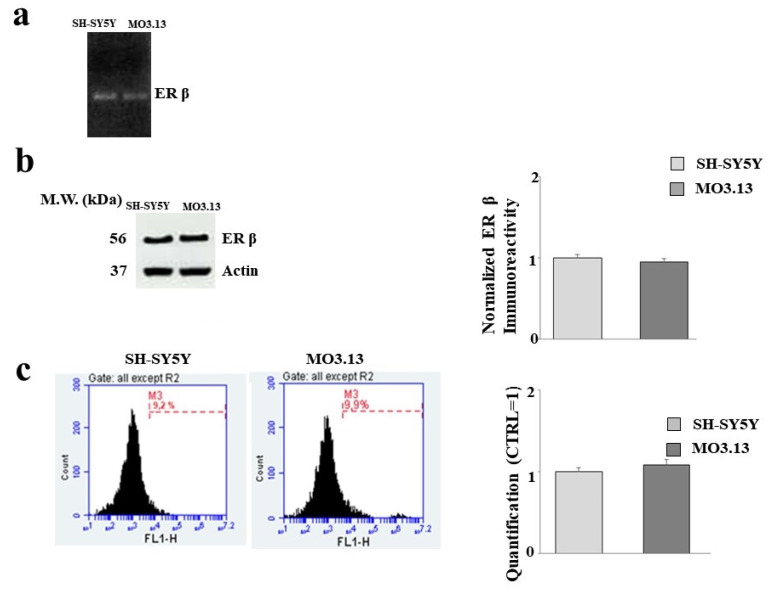
Expression of ER-β. (**a**) The evaluation of genetic information of ER-β by RT-PCR is represented in SH-SY5Y and MO3.13 grown in a co-culture. A representative experiment of three independent experiments (which reported the same results) is shown, and values are expressed as the mean ± standard deviation (sd). The expression of ER-β in both cell lines is represented in panel (**b**), and the resulting values were normalized for the housekeeping protein actin. As shown in panel (**c**), the expression of ER-β was also evaluated by cytometry. For each cytometric reading, 10,000 events were acquired, and in every plot, M3 was used as an arbitrary marker to record variations in fluorescence (a representative experiment from the three independent experiments). Relative quantification was obtained by placing the value obtained for the neurons (1) and comparing the values of the oligodendrocytes. An experiment representative of the three independent experiments that reported the same results is shown. Values are expressed as the mean ± standard deviation (sd). All three methods showed equal levels of ER-β in both cell lines.

**Figure 2 ijms-22-07910-f002:**
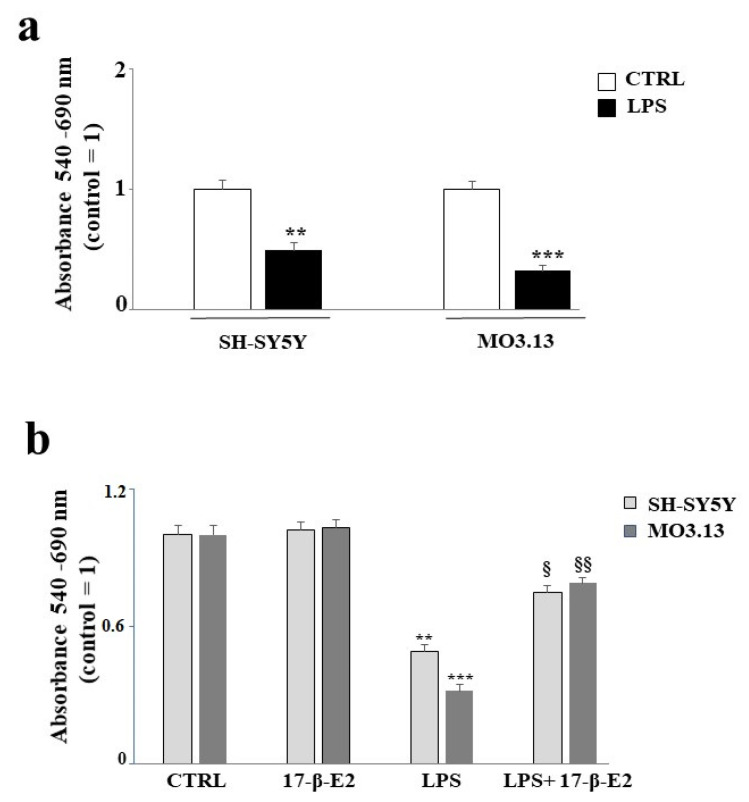
The 17-beta estradiol (17-β-E2) protects oligodendrocytes and neurons from LPS-induced damage. Figure 2 (panel (**a**) shows the extent of the viability reduction induced by the LPS treatment of co-cultured neurons and oligodendrocytes. In particular, the cell lines were treated with LPS at a concentration of 1 μg/mL for 24 h, and at the end of treatment, the cells were subjected to an MTT test. A comparison between neurons and oligodendrocytes under these experimental conditions is shown. (**b**) The results related to co-treatment with 17-β-E2 and LPS. Both cell lines were pretreated with 1 μM 17-β-E2 for 1 h. Without removing or changing the growth medium, the cell lines were exposed to LPS 1 μg/mL for 24 h. Subsequently, cell viability was measured using an MTT assay. Three independent experiments were carried out, with values expressed as the mean ± standard deviation (sd). ** denotes *p* < 0.01 vs. the control; *** denotes *p* < 0.001 vs. the control. § denotes *p* < 0.05 vs. LPS; §§ denotes *p* < 0.01 vs. LPS. Analysis of Variance (ANOVA) was followed by a Tukey–Kramer comparison test.

**Figure 3 ijms-22-07910-f003:**
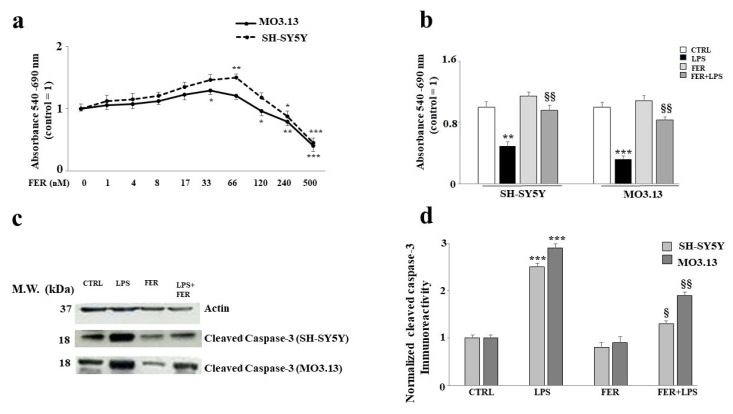
Role of FER in LPS-induced damage. In panel (**a**), a dose–response curve is shown for co-cultured neurons and oligodendrocytes exposed to increasing concentrations of FER over 24 h. The results made it possible to select a 33 nM dose of FER for our experimental model. Three independent experiments were carried out, with the values expressed as the mean ± standard deviation (sd). * denotes *p* < 0.05 vs. the control; ** denotes *p* < 0.01 vs. the control; *** denotes *p* < 0.001 vs. the control. Analysis of Variance (ANOVA) was followed by a Tukey–Kramer comparison test. Panel (**b**) shows the reversal effect of LPS-induced damage by FER, similar to that generated by 17-β-E2. Three independent experiments were carried out, and the values are expressed as the mean ± standard deviation (sd). ** denotes *p* < 0.01 vs. the control; *** denotes *p* < 0.001 vs. the control; §§ denotes *p* < 0.01 vs. LPS. Analysis of Variance (ANOVA) was followed by a Tukey–Kramer comparison test. In panel (**c**), the expression of cleaved caspase 3 in both cell lines (based on the indicated treatments) is displayed. Relative quantification is highlighted in panel (**d**). Three independent experiments were carried out, and the values are expressed as the mean ± standard deviation (sd). *** denotes *p* < 0.001 vs. the control; § denotes *p* < 0.05 vs. LPS; §§ denotes *p* < 0.01 vs. LPS. Analysis of Variance (ANOVA) was followed by a Tukey–Kramer comparison test.

**Figure 4 ijms-22-07910-f004:**
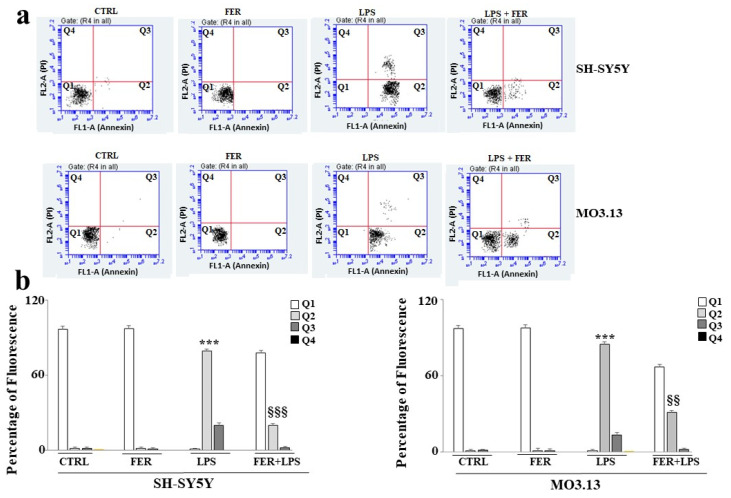
Pretreatment with FER reduces apoptotic death induced by treatment with LPS. To assess whether treatment with LPS resulted in apoptotic or necrotic death and whether FER pretreatment could prevent LPS-induced death, we studied neurons and oligodendrocytes, as shown in Figure 4 (panel (**a**)), where each treatment is represented by a dot plot divided into 4 quadrants (Q1, Q2, Q3, and Q4). Q1 refers to Annexin V-negative/PI-negative cells. Q2 refers to Annexin V-positive/PI-negative cells (apoptosis). Q3 refers to Annexin V-positive/PI-positive cells. Q4 refers to Annexin V-negative/PI-positive cells (advanced necrosis). A representative experiment from the three independent experiments is shown. Panel (**b**)) shows the quantification of the four quadrants of each dot plot for the data obtained from both neurons and oligodendrocytes. Three independent experiments were carried out, and the values are expressed as the mean ± standard deviation (sd). *** denotes *p* < 0.001 vs. the control; §§ denotes *p* < 0.01 vs. LPS; §§§ denotes *p* < 0.001 vs. LPS. Analysis of Variance (ANOVA) was followed by a Tukey–Kramer comparison test.

**Figure 5 ijms-22-07910-f005:**
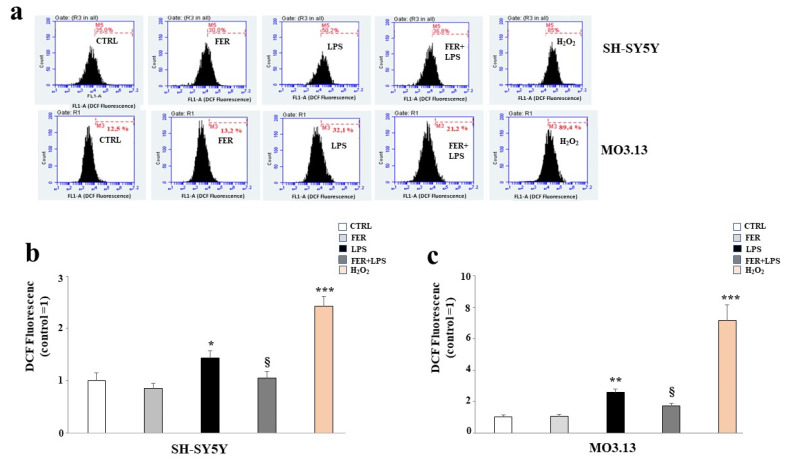
Oxidative profiles in SH-SY5Y and MO3.13 cells: evaluation of ROS accumulation. Co-cultured neurons and oligodendrocytes were treated with LPS or pre-treated with FER and subsequently exposed to LPS, as described. At the end of the treatment, ROS accumulation was assessed by flow cytometry. The results are illustrated in Figure 5. In panel (**a**), the plots for each treatment are displayed as indicated. M5 and M3 are the markers that were arbitrarily designed to determine variations in fluorescence and correspond, respectively, to the fifth and third quantifications carried out. A shift to the right among the cell population indicates an increase in fluorescence and ROS compared to the control, while a shift to the left indicates a reduction in fluorescence and ROS. In both cell lines, H_2_O_2_ was used as the positive control. Panels (**b**,**c**) show the relative quantification obtained by fixing a value for an untreated cell as 1 and comparing the values to those of all other samples. A representative experiment from the three independent experiments that indicates the same results is also displayed. Values are expressed as the mean ± standard deviation (sd). * denotes *p* < 0.05 vs. the control; ** denotes *p* < 0.01 vs. the control; *** denotes *p* < 0.001 vs. the control; § denotes *p* < 0.05 vs. LPS. Analysis of Variance (ANOVA) was followed by a Tukey–Kramer comparison test.

**Figure 6 ijms-22-07910-f006:**
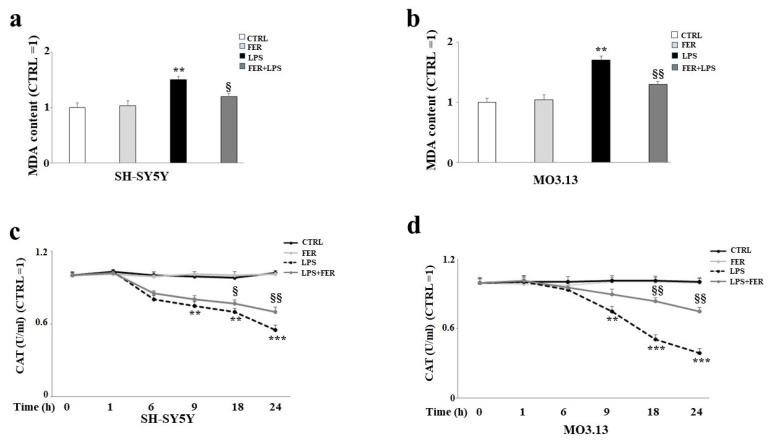
Oxidative profile in SH-SY5Y and MO3.13 cells: evaluation of MDA content and CAT activity. Following treatment with LPS or co-treatment with FER-LPS, the SH-SY5Y and MO3.13 cell lines were subjected to a measurement of their MDA content using a Malondialdehyde assay. The relative quantifications were obtained by fixing a value for an untreated cell as 1 and comparing the values to those of all other samples. The obtained results are shown in Figure 6 (panels (**a**,**b**)). Values are expressed as the mean ± standard deviation (sd). ** denotes *p* < 0.01 vs. the control; § denotes *p* < 0.05 vs. LPS; §§ denotes *p* < 0.01 vs. LPS. Analysis of Variance (ANOVA) was followed by a Tukey–Kramer comparison test. The antioxidative status of the SH-SY5Y and MO3.13 cells, treated as indicated above, was determined by measuring the activity of the antioxidant enzyme catalase. The results are shown in Figure 6 (panels (**c**,**d**)). First, both cell lines were treated with LPS several times (at 1, 6, 9, 18, and 24 h). We then built relative curves for the time-dependent enzymatic activity. Alternatively, we pre-treated the cells with FER and exposed them to LPS for 24 h. The values obtained were normalized based on untreated cells. Values are expressed as the mean ± standard deviation (sd). ** denotes *p* < 0.01 vs. the control; *** denotes *p* < 0.001 vs. the control; § denotes *p* < 0.05.

**Figure 7 ijms-22-07910-f007:**
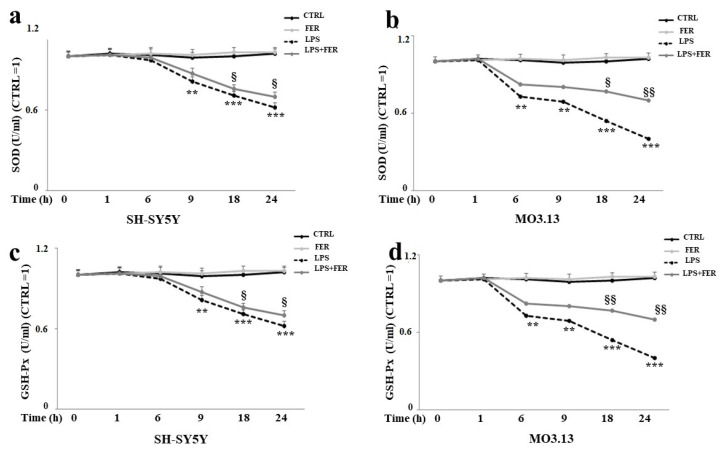
Antioxidative status in SH-SY5Y and MO3.13 cells: evaluation of SOD and GSH-Px activity. The enzymatic activities of SOD and GSH-Px were measured as previously outlined in Figure 6. Panels (**a**,**b**) refer to the measurement of SOD activity, while panels (**c**,**d**) express the results obtained from the measurement of GSH-Px activity. The obtained values were normalized with respect to untreated cells. Values are expressed as the mean ± standard deviation (sd). ** denotes *p* < 0.01 vs. the control; *** denotes *p* < 0.001 vs. the control; § denotes *p* < 0.05 vs. LPS; §§ denotes *p* < 0.01 vs. LPS. Analysis of Variance (ANOVA) was followed by a Tukey–Kramer comparison test.

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
