# Peer review of "The Effect of Ferula communis Extract in Escherichia coli Lipopolysaccharide-Induced Neuroinflammation in Cultured Neurons and Oligodendrocytes"

_ijms, 2021, doi:10.3390/ijms22157910_

Round 1
Reviewer 1 Report
This is a very interesting study concerning the effect of ferutinin on the prevention of cell damage of neuron and oligodendrocyte subjected to LPS stimulation.
However, the in-vitro study did not provide the enough information. The authors should at least conduct the in-vivo study categorized by sham, LPS, and LPS+ferutinin injected to the hippocampus. These animals should receive the neurobehavior assessment such as water maze or Ethovsion and the immunohistochemistry to assess the neuronal survival (TUNNEL) and the associated the inflammatory response such as microglia activation.
Author Response
Dear Reviewer,
thank you for your valuable suggestions. Attached, our answers. The manuscript was also revised in the English language by IJMS .

Reviewer 2 Report
Broad comments
The cell model is simple, but the co-culture setting holds some advantages.
In order to characterize the in vitro model and having a positive control for cell protection to be compared with FER compound of interest, the authors present data including 17-β-E2, which are not novel, but necessary. This is clearly said in the manuscript.
They mainly protection of FER by Ferula communis evaluate apoptosis and oxidative stress.
The manuscript would benefit of a revision before any acceptance, especially in a high quality journal like such one selected for the submission.
Major comments
- The last sentence of the abstract should be reconsidered, since results deriving from LPS damage cannot so simply and directly support the administration to patients with demyelination or neurodegenerative disease, even though the inflammation. LPS is just a model here for the tumor-cell line co-culture, with its great limitations.
- In the figure 1 Legeng there is description for panel a) and panel b), but not for panel c
- In the figure 1 Legend appears “Figure 3”, this is uncorrect and I guess it derives from figure assembling. Please, check it and correct.
- Figure 1 c is not clear at all. Please increase the quality of the picture and precisely describe in figure 1 legend the cytofluorometric results (as described in section 2.1 of Results)
- It seems that Figure 2 panel a) shows results merging the one in panel b) for LPS only condition. I wonder if the authors will want to maintain the panel a) or if this panel will be superfluous.
- SH-SY5Y differentiation was performed with retinoic acid for 5 days, but usually the majority of protocols indicate serum reduction below 10%. On the other hand, the neurons-like differentiated cells switch to 10% serum, a concentration used into the co-colture. Since the SH-SY5Y were also seeded after the retinoic acid protocol, I wonder about the differentiation grade of these cells. Can you please provide evidence of differentiation (gene expression, protein level or mature neural makers) in supplementary information in these conditions, e.g. in 10% serum?
- Bothe the two axis of Figure 4 panel a) should clearly indicate the Annexin and PI label.
- Figure 4 may have some issues. Q4 is always zero event? It’s nice, but after cell manipulation and detachment really strange to do not have any small percentage of necrosis.
- About figure 4 legend. Authors incorrectly repeat a quadrant definition. Indeed, “Q1 refers to Annexin V negative /PI negative cells.” (line 178) and then incorrectly is reported “Q4 refers to Annexin V negative /PI negative cells (advanced necrosis)”. Can they correct? Should be “Q4 refers to Annexin V negative /PI *positive cells” , true?
- Figure 4 a) refers to some “Table 4”. Please be careful. Please remove if there is a mistake or add a specific Table.
- Figure 5 is missing in panel a) the label “DCF fluorescence”. The label addition can help the reader.
- About Figure 5 b and c panels. How is it possible that H2O2 controls are not different form LPS in ROS accumulation (both graphically, e.g. by comparing <3 vs <8 relative quantification, and considering the appearing relative low sd? Can the authors explain this or check the statistics? Thanks
- Figure 5 legend reports on line 206 “M5 and M3 are the markers arbitrarily designed to determine variations in fluorescence.” This is not clear. Please edit the text. Importantly, what are M3 and M5?? Why not use the BD software name for the selected calculated area (shift in population)? And why 3 and 5??
- Was the unstained sample checked for ROS cytofluorometric evaluation and consecutive quantification? It seems the neuronal cell express more ROS basally than the oligodendrocyte cells, but reading specific cell autofluorescence can normalize the quantification. Did the authors preliminary check that?
If it is such case, reconsider Figure 5 results and present revised data. - In Figure 4, b and c) panels, RATIO (%) is not clear considering single Quadrant visualization. Can you make it clearer than now?
- I suggest to improve the legend of Figure 6. Also figure 7 is described there.
- English language can be improved. I suggest a reading of the manuscript by a native English speaker.
Minor comments
- Line 399, “disseminated”. I would suggest of using another word
- Section 4.4. Please mention product code/number, if the antibody is unique and easily identifiable in the company catalog. Ideally this should be included as supplementary file.
- The authors can mention in the discussion the view of valorizing also Ferula communis by-products, considering the potential cytoprotective and anti-inflammatory results they identified.
- . The prospective role and sustainable application of plant derivatives in neuroprotective stategies is evident and can be cited (please have a look at 10.3390/metabo10100408)
- Figure 7 title says “ completion”. I suggest to change word
- In the Title 2.4 I suggest to remove “FITC”
Author Response
Dear Reviewer,
thank you for your valuable suggestions. Attached, our answers. The manuscript was also revised in the English language

Round 2
Reviewer 1 Report
The authors have already addressed the response to the comments with the point to point. The manuscript should be published
Author Response
Thanks for your review.
Jessica Maiuolo

Reviewer 2 Report
The authors kindly and wisely addressed all my comments and improved their manuscript.
Author Response
Thanks for your review.
Jessica Maiuolo
